# Apoptosis-Inducing Factor Deficiency Induces Tissue-Specific Alterations in Autophagy: Insights from a Preclinical Model of Mitochondrial Disease and Exercise Training Effects

**DOI:** 10.3390/antiox11030510

**Published:** 2022-03-07

**Authors:** Sara Laine-Menéndez, Miguel Fernández-de la Torre, Carmen Fiuza-Luces, Aitor Delmiro, Joaquín Arenas, Miguel Ángel Martín, Patricia Boya, Alejandro Lucia, María Morán

**Affiliations:** 1Mitochondrial and Neuromuscular Diseases Laboratory, Instituto de Investigación Sanitaria Hospital ‘12 de Octubre’ (‘imas12’), 28041 Madrid, Spain; sara.laine@cbm.csic.es (S.L.-M.); mftorre@ing.uc3m.es (M.F.-d.l.T.); cfiuza.imas12@h12o.es (C.F.-L.); adelmiro@h12o.es (A.D.); joaquin.arenas@salud.madrid.org (J.A.); mamcasanueva.imas12@h12o.es (M.Á.M.); 2Spanish Network for Biomedical Research in Rare Diseases (CIBERER), U723, 28029 Madrid, Spain; 3Department of Cellular and Molecular Biology, Centro de Investigaciones Biológicas Margarita Salas, CSIC, 28040 Madrid, Spain; pboya@cib.csic.es; 4Faculty of Sports Sciences, European University of Madrid, Villaviciosa de Odón, 28670 Madrid, Spain; alejandro.lucia@universidadeuropea.es; 5Spanish Network for Biomedical Research in Fragility and Healthy Aging (CIBERFES), 28029 Madrid, Spain

**Keywords:** autophagy, OXPHOS, mitochondrial diseases, Harlequin, heart, skeletal muscle, cerebellum, brain

## Abstract

We analyzed the effects of apoptosis-inducing factor (AIF) deficiency, as well as those of an exercise training intervention on autophagy across tissues (heart, skeletal muscle, cerebellum and brain), that are primarily affected by mitochondrial diseases, using a preclinical model of these conditions, the Harlequin (Hq) mouse. Autophagy markers were analyzed in: (i) 2, 3 and 6 month-old male wild-type (WT) and Hq mice, and (ii) WT and Hq male mice that were allocated to an exercise training or sedentary group. The exercise training started upon onset of the first symptoms of ataxia in Hq mice and lasted for 8 weeks. Higher content of autophagy markers and free amino acids, and lower levels of sarcomeric proteins were found in the skeletal muscle and heart of Hq mice, suggesting increased protein catabolism. Leupeptin-treatment demonstrated normal autophagic flux in the Hq heart and the absence of mitophagy. In the cerebellum and brain, a lower abundance of Beclin 1 and ATG16L was detected, whereas higher levels of the autophagy substrate p62 and LAMP1 levels were observed in the cerebellum. The exercise intervention did not counteract the autophagy alterations found in any of the analyzed tissues. In conclusion, AIF deficiency induces tissue-specific alteration of autophagy in the Hq mouse, with accumulation of autophagy markers and free amino acids in the heart and skeletal muscle, but lower levels of autophagy-related proteins in the cerebellum and brain. Exercise intervention, at least if starting when muscle atrophy and neurological symptoms are already present, is not sufficient to mitigate autophagy perturbations.

## 1. Introduction

Mitochondrial diseases (MD) are caused by mutations in mitochondrial or nuclear genes with subsequent impairment of the oxidative phosphorylation system (OXPHOS). Despite their low prevalence (~1 in 5000 individuals), MD represent the most frequent inborn errors of metabolism [1]. There is individual variability in symptom onset and clinical manifestations and, although virtually any tissue can be affected skeletal muscle, the heart and central nervous system (CNS) are the primary disease targets [2].

At the cellular level, OXPHOS dysfunction leads to impairments in ATP production, mitochondrial membrane potential and the assembly of OXPHOS complexes, together with an increase in reactive oxygen species (ROS) production, thereby leading to ‘energy crises’, oxidative stress and cellular damage [3,4]. Alterations in autophagy—the lysosomal-dependent process that recycles damaged or dysfunctional cellular components including mitochondria (i.e., ‘mitophagy’)—can also occur in MD; this is reflected by an increase in autophagy-related vesicles in transmitochondrial cybrids harboring pathogenic mitochondrial DNA mutations, as well as in patient-derived fibroblasts or in tissues from preclinical models [5,6,7,8,9,10,11,12,13,14,15,16,17]. However, the actual relevance of autophagy alterations in the pathophysiology of MD remains to be fully elucidated and, it is in fact unknown whether the different affected tissues show this type of alterations.

There is essentially no effective cure for most MD, although some strategies can help to attenuate clinical manifestations [18]. One such strategies is regular physical exercise, which has proven to mitigate exercise intolerance in affected patients [19,20,21,22,23,24,25]. However, besides improvements in muscle oxidative capacity, the molecular bases underlying exercise benefits are not yet fully clear. In this regard, there is growing evidence that regular exercise produces beneficial effects at the multisystemic level through numerous molecular pathways, including the promotion of autophagy—especially in those organs that are primarily affected in MD, that is, skeletal muscle, the heart and CNS [26,27,28,29,30]. Furthermore, exercise has been reported to attenuate dysregulation of autophagy in several conditions associated with altered autophagy, such as age-related sarcopenia, spinal muscular atrophy, hypertension or Parkinson’s disease [31,32,33,34]. However, it remains to be determined whether exercise can improve autophagy in the context of MD.

Apoptosis inducing factor (AIF) is a mitochondrial protein that was first reported to induce chromatin degradation and caspase-independent cell death by migrating to the nucleus in response to an apoptogenic insult [35,36,37]. More recently, it has been elucidated that AIF cooperates with the mitochondrial protein MIA 40 in the import of some subunits of the respiratory complex I and of complex IV assembly factors through a disulfide relay substrate oxidation mechanism [38,39]. The deficiency of AIF results in altered assembly and low activity of the respiratory chain complex I in preclinical and clinical models [40]. The aim of the present study was to analyze the status of autophagy in the affected tissues of a well-characterized preclinical model of MD, the Harlequin (Hq) mouse. This animal exhibits a decrease of ~80% in the apoptosis inducing factor (*Aif*) gene expression due to a proviral insertion in this gene. Thus, the Hq model recapitulates important patients’ alterations due to respiratory chain complex I deficiency, such as myopathy, increased risk of cardiomyopathy, and cerebellar ataxia [41,42,43,44,45]. In addition, we analyzed whether a physical exercise intervention combining aerobic and resistance training modalities could promote changes in autophagy in the aforementioned affected tissues. Our results showed a tissue-specific response, with increased autophagy in the heart and *biceps femoris* of the Hq mouse, and no significant training effect on autophagy in any of the analyzed tissues.

## 2. Materials and Methods

### 2.1. Mouse Model

All experimental protocols were approved by the Institutional Review Committee (*Hospital 12 de Octubre*; project numbers PROEX 111/15 and PROEX 067/18) and were conducted in accordance with Animal Research: Reporting of In Vivo Experiments (ARRIVE) guidelines and with European (European Convention for the Protection of Vertebrate Animals ETS123) and Spanish (32/2007 and R.D. 1201/2005) laws on animal protection in research. Heterozygous male (B6CBACa Aw-J/A-Aifm1Hq/J) and female Hq mice (X/Hq, B6CBACa Aw-J/A-Aifm1Hq/J), and wild-type male mice (WT, B6CBACa) of the same strain, were obtained from The Jackson Laboratory (Bar Harbor, ME, USA) were used for the present study. Mice were housed in the animal facility of *Hospital 12 de Octubre* (Madrid, Spain) under controlled conditions of temperature, humidity and ventilation, with 12-h light/dark cycles and ad libitum access to food and water.

### 2.2. Study Design

#### 2.2.1. Time Course Study

To assess the impact of MD progression on autophagy, 8-week-old heterozygous female mice (X/Hq, B6CBACa Aw-J/A-Aifm1Hq/J) were crossed with WT males of the same strain (all purchased from The Jackson Laboratory), and the male mice of the F1 generation were used for tissue analyses at different ages. PCR genotyping of F1 mice was performed on tail DNA using previously described primers (Integrated DNA Technologies; Coralville, IA, USA) [43].

F1 male mice were sacrificed at 2, 3 and 6 months of age (8 WT and 8 Hq mice for each age, project number PROEX 067/18) by cervical dislocation, and the heart, *biceps femoris*, *quadriceps femoris*, brain and cerebellum were dissected, frozen in liquid nitrogen and stored at −80 °C for subsequent biochemical analysis.

#### 2.2.2. Exercise Training Study

The study variables were analyzed in a group of mice after an 8-week exercise training program previously described by us [41]. Briefly, 44 male mice (23 WT and 21 Hq, project number PROEX 111/15) were obtained from The Jackson Laboratory from 6–8 weeks of age and were physically evaluated every week until the onset of MD symptoms. Physical evaluation included locomotion analysis (treadmill locomotion test), as well as cerebellar ataxia (rotarod) and muscular strength (handgrip) determinations. When symptoms of MD were clear, each Hq mouse was paired to a WT mouse; both mice then performed a maximal incremental test on a treadmill and were randomly allocated to a sedentary (Sed, 12 WT and 11 Hq) or an exercise training group (Ex, 11 WT and 11 Hq). In the latter, each training session lasted between 40 and 60 min, and included resistance exercises thrice a week (Monday, Wednesday and Friday), and endurance (treadmill) training five times per week (from Monday to Friday), with the duration and intensity of each session gradually increasing during the program. Sedentary animals did not perform the training program but were free to move in their cages. The mean ± SD age of the Hq mice upon symptom onset and at the end of the intervention period were 12.1 ± 0.1 weeks and 5.3 ± 0.1 months, respectively. The mean age of the mice at the end of the intervention period did not differ between the experimental groups [41].

After the 8-week intervention study, the aforementioned physical assessments were also performed in age-matched pairs of sedentary and exercise-trained mice. In order to account for a potential confounding effect caused by the short-term response to a single acute exercise stimulus—we solely intended to determine the long-term (i.e., ‘training’) effects of the exercise intervention—48 h elapsed after the last exercise test before mice were sacrificed by cervical dislocation. The heart, *biceps femoris*, *quadriceps femoris*, brain and cerebellum were quickly obtained and frozen in liquid nitrogen before storage at −80 °C until molecular analysis.

### 2.3. Cardiomyocyte Cell Culture

Cardiomyocyte isolation was performed using the PierceTM Primary Cardiomyocyte Isolation Kit (Thermo Fisher Scientific; Waltham, MA, USA) following the manufacturer’s instructions. Briefly, hearts from postnatal (1–4 days old, P1–P4) mice were dissected, cut into 1–3 mm fragments, and washed twice with Hanks’ Balanced Salt Solution without calcium and magnesium at 4 °C. Tissue fragments were subsequently digested by enzymatic treatment, mechanically disaggregated, and living cells were counted for cell culture seeding. Cells were cultured in Dulbecco’s Modified Eagle Medium (DMEM) (Gibco; Gaithersburg, MD, USA), supplemented with 10% fetal bovine serum (Gibco), 100 IU/mL penicillin, 100 IU/mL streptomycin and the cardiomyocyte growth factor provided by Pierce in the isolation kit (1/1000), in a CO_2_ incubator at 37 °C. The culture medium was replaced every 3 days, and the obtained cardiomyocyte purity was approximately 80%.

### 2.4. Assessment of Autophagy Flux

For the study of autophagic flux, 3-month-old male mice (5 WT and 5 Hq) were treated i.p., with 0.5 mL of sterile phosphate buffered saline (PBS) containing 40 mg/kg leupeptin hemisulfate (Merck; Darmstadt, Germany). After 30 min treatment, mice were sacrificed by cervical dislocation and tissues were dissected, frozen and stored as described above.

To assess autophagic flux in cultured cardiomyocytes, cells were treated with 30 µg/µL of hydroxychloroquine sulfate (H0915, Sigma-Aldrich; St., Louis, MI, USA) for 24 h (5–15 cells of 2–3 mice). After treatment, cells were fixed and processed for immunofluorescence as described below.

### 2.5. Tissue Processing

Heart and muscle tissues were processed to obtain total homogenates in ice-cold 10 mM of Tris-HCl pH 7.6, containing 150 mM of NaCl, 1 mM of EDTA, 1% Triton™ X-100 and a protease and phosphatase inhibitor cocktail (Roche Diagnostics Corporation; Indianapolis, IN, USA) 1:10 (weight: volume) in a Potter homogenizer. The brain and cerebellum were homogenized at 1:10 (weight: volume) in ice-cold RIPA buffer containing 50 mM of Tris-HCl pH 7.4, 1% NP-40, 0.5% Na-deoxycholate, 1% SDS, 150 mM of NaCl, 2 mM of EDTA, with protease and phosphatase inhibitors. Homogenates were centrifuged at 11,000× *g* for 15 min at 4 °C, and the supernatants containing the solubilized proteins were collected for analysis. Protein concentration in the homogenates was determined using the Pierce^®^ BCA protein assay kit (Thermo Fisher Scientific; Whaltham, MA, USA).

### 2.6. Western Blotting

Autophagy-related protein levels were analyzed by Western blotting. Samples of tissue homogenates containing between 20 and 40 µg of denatured proteins were separated on 7.5–15% SDS-polyacrylamide gels. Resolved proteins were transferred to PVDF membranes and blocked with 5% non-fat dry milk or 5% bovine serum albumin (BSA). Blocked membranes were probed with the relevant primary antibody (Appendix A) and subsequently incubated with horseradish peroxidase (HRP)-conjugated secondary antibody. Protein signals were finally detected with ECL Prime Western blotting Detection Reagent (Amersham, GE Healthcare; Little Chalfont, UK). Protein signals were quantified with ImageJ (Rasband, W.S, ImageJ; U. S. National Institutes of Health, Bethesda, MA, USA). Data were normalized by total protein load per well according to Coomassie Blue staining of the membrane as previously described [46] for heart protein analysis, whereas glyceraldehyde 3-phosphate dehydrogenase (GAPDH) was used as the loading control for skeletal muscle analyses and γ-tubulin for brain and cerebellum studies.

### 2.7. Immunocytochemistry

Cardiomyocytes were seeded and fixed on glass coverslips with 4% paraformaldehyde in a CO_2_ incubator. Fixed cells were washed with PBS and antigen retrieval was performed with 5% urea and 0.1 mM Tris pH 9.5 for 20 min at 95 °C. Cells were permeabilized with 0.1% Triton™ X-100, blocked with 10% goat serum and 0.01% Triton X-100 in PBS, and incubated with primary antibody in blocking solution overnight (Appendix A). After washing, cardiomyocytes were incubated with the relevant fluorophore-coupled secondary antibody in blocking solution. After secondary antibody washing, nuclei were stained with 0.5 µg/mL of 4′,6-diaminidino-2-phenylindole (DAPI) and mounted in Prolong Gold Antifade mounting medium (Thermo Scientific; Whaltham, MA, USA). Cells were observed with an LSM510 META confocal microscope (Zeiss; Oberkochen, Germany) fitted with a 40× achromatic or 63× plan-achromatic objective.

### 2.8. High Performance Liquid Chromatography

The amino acid content of heart and skeletal muscle homogenates was analyzed by high-performance liquid chromatography (HPLC) using a methodology certified by the Spanish National Accreditation Entity ENAC (*Entidad Nacional de Acreditación*) (ISO15189) used for the analysis of clinical samples in the Clinical Biochemistry Department of the *Hospital 12 de Octubre*. Briefly, homogenate samples containing 1.7 and 2.0 mg of total protein for heart and skeletal muscles homogenates, respectively, were deproteinized with 50% sulfosalicylic acid and analyzed by ion exchange chromatography with post-column derivatization with ninhydrin on a Biochrom 30+ amino acid analyzer (Biochrom Ltd.; Cambridge, UK). Amino acid peaks were detected and quantified with OpenLAB EZChrom Edition software A.04.10 (Siegwerk Druckfarben AG & Co. KGaA; Siegburg, Germany).

### 2.9. Statistical Analysis

All study variables were expressed as a median and interquartile range. All variables were tested for normality of data distribution using the D’Agostino-Pearson test. Owing to the relatively small sample size, and to the fact that most of the study variables did not follow a Gaussian (normal) distribution, the non-parametric Kruskal–Wallis test was used to determine whether a significant ‘group’ effect was found for the different variables, in which case pairwise comparisons were performed *post hoc* with the Dunn’s test. The Mann–Whitney U test was used for comparisons between pairs of groups. Statistical significance was set at a *p*-value *p* < 0.05. All analyses were performed with GraphPad Prism^®^ 7 for Windows (GraphPad Software; San Diego, CA, USA).

## 3. Results

### 3.1. Autophagy in the Heart: Time Course Study

To analyze if MD was associated with autophagy alterations depending on age, the levels of proteins representative of distinct phases of the autophagy pathway were measured at 2, 3 and 6 months of age, respectively, in WT and Hq mice of the time-course study (Figure 1). The analysis of the main regulators of autophagy showed the following results. First, the levels of the active form (phosphorylated at residue Ser2448) of the mechanistic target of rapamycin (pmTOR) tended to be higher in Hq mice than in WT mice at all ages, with the difference between both groups significant at 3 months of age (Figure 1). In turn, the levels of 5′ AMP-activated protein kinase (AMPK) phosphorylated at residue Thr172 (pAMPK), which is associated with a state of autophagy activation in response to low cellular energy availability, tended to be lower in Hq mice than in WT, with the between-group difference significant at 2 months of age. Subsequently, the levels of two proteins involved in the early stages of formation of the phagophore (i.e., the double membrane that encloses and isolates the cytoplasmic components during autophagy), Beclin 1 and autophagy-related protein 16-like (ATG16L), were quantified. We found the trend towards higher levels of both proteins in Hq mice compared to WT mice, with the difference between groups significant at 2 (for both Beclin 1 and ATG16L) and 6 months of age (for ATG16L only). The determination of microtubule-associated protein 1B light chain I (LC3B)-I and LC3B-II (which is formed after LC3B-I conjugation to phosphatidylethanolamine) showed that LC3B-I levels tended to be higher in Hq mice at all ages, with the between-group difference significant at 2 and 3 months. Next, we measured the levels of the protein p62, which tended to be higher in Hq mice than in WT mice at all ages, with between-group differences significant at 2 and 3 months. Finally, to assess the lysosomal content in Hq mice, the levels of the lysosomal protease cathepsin B (protein bands at 27 and 24 KDa) were quantified, with the results showing a trend towards higher values in Hq mice and with a significant difference vs. WT mice for the 27 KDa band at 2 months of age (Figure 1).

### 3.2. Autophagic Flux Analysis in the Heart

To assess whether the accumulation of autophagy proteins in the hearts of Hq mice was due to a failure in the autophagosome-lysosome flux or to an autophagy induction response, we studied the autophagic flux by treating 3-month-old WT and Hq animals with leupeptin, an inhibitor of cysteine, serine and threonine peptidases present in lysosomes. The levels of different autophagy proteins in leupeptin-treated mice are shown in Figure 2a. We found a significant group effect for the levels of Beclin 1, p62 and LC3B-II (*p* = 0.0038, *p* = 0.014 and *p* = 0.015, respectively), with higher values of the three variables in treated mice than in untreated animals (Figure 2a). *Post hoc* analyses showed significant differences in Beclin 1 and p62 levels between leupeptin-treated Hq and non-treated WT mice as well as in p62 levels between treated and non-treated Hq, and a similar response to leupeptin in Hq and WT mice with overall higher autophagy markers for treated mice (both genotypes). These data suggest a normal autophagic flux in the heart of both Hq and WT mice and may point to an increased protein degradation response by the lysosome-autophagosome pathway in Hq mice.

To assess whether the higher protein tagging for autophagic degradation in the hearts of Hq animals was due to a higher lysosomal degradation of mitochondrial proteins, or perhaps to mitophagy, we quantified the levels of several representative mitochondrial proteins in leupeptin-treated and non-treated mice. We found no significant differences in the levels of mitochondrial NADH dehydrogenase (ubiquinone) 1 beta subcomplex subunit 8 (NDUFB8), the voltage-dependent anion-selective channel protein 1 (VDAC1), or the mitochondrial import receptor subunit TOM20, respectively, between leupeptin-treated and untreated Hq mice. However, we detected a significant group effect for the ATP synthase lipid-binding protein (ATP5A) (*p* = 0.0026), which reached higher levels in leupeptin-treated animals with significant *post hoc* differences observed for leupeptin-treated Hq mice compared to both untreated Hq and WT animals (whether treated or not) mice (Figure 2). These results suggest that, overall, mitophagy is not enhanced in the heart of Hq mice and that autophagy participates in the turnover of some OXPHOS proteins.

Finally, we detected LC3B by immunofluorescence in WT and Hq cardiomyocytes isolated from neonatal mice treated with hydroxychloroquine (a lysosomal inhibitor which basifies lysosomes) for 24 h (Figure 2b,c). The analysis of LC3B fluorescence intensity per cell showed significant differences between groups (*p* = 0.036), although *post hoc* tests did not reveal differences between pairs of groups. A higher LC3B fluorescence was observed in hydroxychloroquine-treated Hq cardiomyocytes compared to untreated Hq cells, with this difference greater than that observed between treated and untreated WT cardiomyocytes. These findings may suggest a more active autophagic flux in neonatal Hq cardiomyocytes compared to WT cells (Figure 2c).

### 3.3. Effects of Exercise Training on Autophagy in the Heart

In order to determine if exercise training could induce normalization of autophagy markers in the heart tissue of Hq mice, we measured several autophagy-related proteins in sedentary and exercise-trained animals (Figure 3). We found significant differences between experimental groups for most variables (pmTOR, *p* = 0.0008; Beclin 1 and ATG16L *p* < 0.0001; LC3B-I, *p* = 0.015; LC3B-II, *p* = 0.0021; p62, *p* < 0.0001; and cathepsin B (*p* = 0.002 for 27KDa and *p* = 0.0305 for 24 KDa band)) (Figure 3). However, *post hoc* analyses revealed that the training intervention did not induce a significant effect in either Hq or WT. All the observed differences between pairs of experimental groups were due to genotype (i.e., MD) and corroborated our previous results regarding the differences in ATG16L and Beclin 1 levels between WT and sedentary Hq mice. In addition, statistical significance was also reached for *post hoc* differences in LC3B-I and II, pmTOR, p62 and cathepsin B (27 KDa) between sedentary Hq and WT mice. We found quasi-significant differences in pAMPK between sedentary Hq and WT mice (*p* = 0.059) (Figure 3).

### 3.4. Sarcomere Proteins Levels in the Heart

To analyze if autophagy might be involved in the degradation of the contractile machinery of the heart, we measured the levels of three sarcomere proteins in the hearts of mice in the intervention study. The results are shown in Figure 4. We found a significant group effect for the proteins myosin light chain 3 (MYL3), cardiac troponin I3 (TNNI3) and cardiac troponin C1 (TNNC1) (*p*-value = 0.0006, 0.0002 and 0.0071, respectively), with overall lower levels of these proteins in Hq mice (Figure 4). In *post hoc* pairwise comparisons, sedentary Hq mice showed lower levels of TNNI3 and TNNC1 than sedentary WT mice (Figure 4). On the other hand, the exercise training intervention did not induce any significant change (i.e., no differences between exercise-trained and sedentary Hq, respectively). However, strikingly, we found lower levels of TNNC1 in trained WT mice than in sedentary WT mice.

### 3.5. Free Amino Acid and Ammonium Levels in the Heart

To determine whether the increased autophagic flux in Hq mice might be related to increased protein catabolism, we measured the concentration of free amino acids by HPLC in total heart homogenates from sedentary WT and Hq mice of the intervention study groups. We found a trend towards higher levels of several amino acids in Hq mice compared to WT mice, with significant differences for threonine, serine, glycine and valine (Table 1). Ammonium levels were similar in both groups (*p* > 0.99).

### 3.6. Autophagy in Skeletal Muscle—Exercise Training Effects

To study whether autophagy was also altered in skeletal muscle, as well as the potential effects of exercise training in this process, we measured the levels of different proteins representing distinct phases of autophagy in homogenates of the *biceps femoris* of the exercise training study groups (Figure 5). We found a significant group effect for total AMPK (tAMPK) levels (*p* = 0.025), with *post hoc* analysis revealing higher tAMPK in both Hq mice (exercise-trained or sedentary) than in sedentary WT mice. However, the analysis of pAMPK levels showed no difference between groups, indicating that the training intervention did not produce a significant effect on total or phosphorylated AMPK levels in any of the experimental groups. We also detected a significant group effect for Beclin 1, ATG16L, LC3B-II and p62 (*p* = 0.004, *p* < 0.001, *p* < 0.001 and *p* = 0.0046, respectively), with *post hoc* comparisons showing higher levels of Beclin 1, ATG16L and LC3B-II in both Hq groups than in sedentary WT animals, as well as higher levels of these proteins in sedentary Hq mice compared to sedentary WT mice. In the case of p62, the post hoc test only showed differences between sedentary groups of mice. On the other hand, we did not observe a significant group effect for LC3B-I or lysosomal-associated membrane protein 1 (LAMP1) levels between experimental groups, for either disease or training effect (Figure 5).

### 3.7. Free Amino Acids and Ammonium Levels in Skeletal Muscle

To determine whether an increase in protein catabolism status might also occur in the skeletal muscles of Hq mice, we quantified the concentration of free amino acids and ammonium by HPLC in total quadricep homogenates from WT and Hq sedentary mice from the exercise training study. We found a trend towards higher levels of virtually all amino acids in Hq mice compared to WT mice, with significant differences for valine, ornithine, lysine and arginine (Table 2). By contrast, we detected no differences in ammonium levels between Hq and WT mice.

### 3.8. Autophagy Protein Levels and Effects of Exercise Training in Central Nervous System

To assess whether autophagy was altered in the CNS, as well as the potential effects of exercise training on autophagy markers in this tissue, we measured several autophagy-related proteins in the cerebellum and brain of sedentary and exercise-trained mice (Figure 6). We found a significant effect for the levels of Beclin 1 and ATG16L in the cerebellum (*p* = 0.0004 and *p* = 0.0032, respectively) and brain (*p* = 0.0007 and *p* = 0.0003, respectively) (Figure 6a,b). The *post hoc* analysis revealed only significant differences between both groups of Hq mice (trained or sedentary) and WT sedentary mice, and no effect of the exercise training intervention in WT or Hq mice (Figure 6a,b). Regarding LC3B-I and LC3B-II, we found no differences in the cerebellum or brain, indicating the absence of autophagosome accumulation in the CNS of Hq mice and no effects of the exercise training program on autophagosome content (Figure 6a,b). By contrast, we detected a higher content of p62 and LAMP1 (*p* = 0.0084; *p* = 0.0053, respectively) in the cerebellum of Hq mice but not in the brain. *Post hoc* analyses revealed significant differences only in sedentary Hq mice in comparison with trained WT mice for p62, and differences between the exercise-trained groups of Hq and WT mice for LAMP1, but no effect of the exercise intervention in WT and Hq mice (Figure 6a).

Finally, the analysis of leupeptin treatment of WT and Hq mice (Figure 7) showed significant accumulation of LC3B-I and LC3B-II in both WT and Hq leupeptin-treated cerebella (*p* = 0.0028 and *p* = 0.0067, respectively) and a significant *post hoc* difference between treated Hq and non-treated WT, revealing normal autophagosome elimination in this tissue in the Hq mouse (Figure 7). Protein p62 was also apparently higher in WT and Hq mouse leupeptin-treated cerebella, supporting normal autophagic flux in the Hq cerebellum. However, we found no significant group effect, probably due to the variability in the low number of animals analyzed (*n* = 5–6 in each experimental group), and the time selected for the treatment, which was optimal for the heart but not for the CNS [47].

## 4. Discussion

The present study suggests that AIF deficiency causes a tissue-specific effect on autophagy, with the accumulation of autophagy-related proteins and a higher content of free amino acids in the heart and skeletal muscle tissue, as well as increased autophagy flux and lower levels of sarcomeric proteins in the heart, but, in turn, with the attenuation of autophagy in the CNS. On the other hand, an exercise intervention combining endurance and resistance training did not significantly mitigate autophagy alterations in the tissues that are primarily affected by MD (i.e., skeletal muscle, the heart and CNS).

In the present work, the finding of higher levels of proteins involved in different stages of autophagy in the heart of Hq mice suggest either an enhancement of autophagy, or an alteration of the autophagic flux due to AIF deficiency. However, the fact that treatment with leupeptin (a blocker of autophagy) in Hq mice demonstrated normal autophagic flux in the heart, points to a higher autophagy flux in this tissue as a result of AIF deficiency. Although we did not assess autophagy flux in skeletal muscles of Hq mice, the accumulation of autophagy markers observed in this tissue would suggest the existence of a similar response to that found in heart and correlates with a higher p62 and LC3II/I ratio previously reported in the Deletor mouse model of MD [48]. In contrast, the brain and the cerebellum did not show accumulation of autophagy markers in Hq mice but, in fact, a slightly lower content of Beclin 1 and ATG16L, and according to LC3B-II levels, a very low content of autophagosomes. In this regard, previous research has demonstrated a negative effect of Beclin 1 deficiency on Purkinje cell survival in mice [49] and that loss of ATG proteins in neurons leads to motor dyscoordination and progressive neurodegeneration [50,51], both of which are hallmarks of the Hq mice phenotype [42,43,52]. Considering that leupeptin treatment revealed normal autophagosome-lysosome flux in the cerebellum, our results might indicate a lower induction of autophagy in the CNS of Hq mice, which in the cerebellum could be responsible for the accumulation of p62-linked proteins. This high p62 content may also point to additional roles of p62 in the cerebellum of Hq mice, such as compensatory proteasome-mediated protein degradation [53,54], antioxidant response mediated by the Kelch-like ECH-associated protein (Keap1)/nuclear factor erythroid-derived 2-like 2 (Nrf2) pathway [55,56] or apoptosis [57]. The higher LAMP1 content found in the cerebellum indicates a higher lysosomal content in the Hq mouse, which can hypothetically be attributed to lysosomal biogenesis to promote a compensatory degradation of cellular components by direct lysosomal degradation, such as chaperone-mediated autophagy and/or microautophagy [58,59]. We analyzed if the enhanced autophagy observed in the Hq mouse heart could be related to mitophagy. In this effect, we found that the levels of markers for different mitochondrial compartments (inner and outer membrane) were not sensitive to leupeptin treatment, thereby suggesting that AIF deficiency does not promote mitochondrial degradation by mitophagy. Nevertheless, our finding of a higher content of ATP5A in the heart of leupeptin-treated mice compared with untreated mice would indicate that this mitochondrial protein is, indeed, degraded by the lysosomal pathway, which is consistent with the results of a previous study reporting that autophagy acts selectively on some mitochondrial proteins [60]. Overall, our findings indicate that the alterations in autophagy induced by AIF deficiency in the Hq mouse model are tissue-specific, with an induction of autophagy in striated muscles, but an attenuation in the CNS. Future studies are needed on the role of autophagy alterations in this mouse model of MD.

One of the mTOR-independent signals capable of inducing autophagy is oxidative stress [59], which is also involved in the physiopathology of MD. Indeed, other authors [43] as well as our group [41], have previously reported evidence of oxidative stress in the skeletal muscle of the Hq mouse together with higher protein and activity levels of antioxidant enzymes [41,43]. In addition, a higher sensitivity to oxidative stress-induced dead has been reported in the myocardium of these animals [45], and in the present study we found a higher content of peroxiredoxin 6 and a lower catalase activity in the hearts of 6-month-old Hq mice (data not shown). Therefore, we could hypothesize that oxidative stress associated to AIF deficiency could induce autophagy in the striated (skeletal and heart) muscles of Hq mice. There are also evidences of oxidative stress and enhanced ROS sensitivity in the cerebellum of the Hq mouse model [42,52]. Therefore, the aforementioned findings together with the fact that we failed to detect an increased autophagy in the CNS of Hq mice would suggest that the autophagic response to oxidative stress and mitochondrial dysfunction is also tissue-specific in this preclinical model of MD.

Autophagy is a cellular stress response that can be induced by exercise, not only in skeletal muscles but also in other tissues such as the brain, adipose tissue, pancreas or liver [27,28,30,61,62,63,64]. Interestingly, exercise intervention has been reported to mitigate autophagy alterations in several conditions such as muscle atrophy, heart failure, ischemic stroke, Parkinson’s and Alzheimer’s disease or ataxia [31,65,66,67,68,69]. In the present study, however, despite the significant exercise training-induced improvement in strength and endurance capacity previously reported by us in the same Hq mice of the present study [41], we failed to detect an attenuation of autophagy alterations in the striated muscles or the CNS of these animals with the exercise intervention. In this regard, previous research assessing the short-term effects of an acute exercise session has suggested that high-intensity exercise sessions induce autophagy to a greater extent than longer but less intense ones [63,70]. In addition, autophagy induction can be detected immediately after acute exercise in skeletal muscles, but its persistence thereafter has been questioned [71]. Thus, the mild intensity of the training program used in the present work, together with the fact that animals were sacrificed 48 h after the last physical test to prevent a potential confounding effect produced by an acute exercise stimulus, might explain, at least partly, the lack of changes in autophagy with the exercise training intervention. In addition, an earlier start of the intervention—before symptom onset−as well the use of a longer program could have also normalized the observed autophagy alterations and perhaps increased the lifespan of the animals. In fact, longer exercise interventions have been shown to attenuate proteome deregulation in the skeletal muscle and CNS of the Mutator mouse model of MD [72] or mitigate myopathy and increase the lifespan of the cytochrome C oxidase assembly factor heme A: farnesyltransferase (COX10) knockout (Cox 10^−/−^) mouse model [73]. The fact that our intervention started when myopathy and ataxia were already present might also explain, at least partly, the absence of attenuations in the observed autophagy alterations with exercise training. In this regard, it has been reported that the response of autophagy to exercise can be attenuated is some metabolic diseases [74]. In turn, previous research has indicated a tissue-specific effect of exercise training on autophagy response. For instance, Sprague Dawley rats that underwent a long-term, moderate intensity exercise training program showed autophagy induction in the brain cortex but not in the hippocampus, heart or skeletal muscle [61]. In other studies, acute (i.e., a single bout) as well as long-term exercise (i.e., training) in mice increased autophagy in the brain cortex but not in the cerebellum and hippocampus [30,75]. Therefore, we cannot rule out the possibility that, despite its moderate intensity, the training program used in our study may have elicited a significant autophagic response in other skeletal muscles or brain areas (e.g., brain cortex) that were not analyzed here.

The study of proteins involved in autophagy regulation in the heart showed lower levels of activated AMPK in Hq mice and a higher activation of the autophagy inhibitor mTOR [59]. The absence of activation of AMPK in the heart of Hq mice might be explained by the absence of OXPHOS deficiency previously described in this organ [44]. On the other hand, the enhanced mTOR activation in the Hq heart is in agreement with previous observations of increased mTOR activity in other mouse models of MD, such as the Aifm1 (R200 del) model of AIF deficiency and the Deletor mouse [48,76,77,78]. In these two preclinical models, mTOR overactivation was related to an enhancement of folate-mediated 1-carbon metabolism [48,76]. This metabolic route supplies glycine, serine and folate needed for methylation of transfer RNA during the translation of mitochondrial proteins and, therefore, supports the maintenance of the OXPHOS system [79]. Moreover, 1-carbon metabolism has been reported to facilitate the assembly of respiratory chain complex I also by a mechanism mediated by serine catabolism [58]. Therefore, overactivation of mTOR may act as a compensatory mechanism to promote the 1-carbon metabolism in the Hq mouse heart. In this effect, in the Deletor mouse, 1-carbon metabolism enhancement is associated with higher free amino acid content in the skeletal muscle and heart and accumulation of autophagic markers in the former [48,77]. In agreement with these findings, we observed higher serine and glycine content together with higher mTOR activation in the hearts of Hq mice than in those of the control mice. Therefore, one could hypothesize that folate-mediated 1-carbon metabolism is promoted by mTOR activation as a compensatory mechanism to enhance respiratory chain complex I and OXPHOS function in the Hq mouse heart. Under these conditions, autophagy would be driven by an mTOR-independent signal [59], as a synergistic response that induces protein catabolism and amino acid release to feed 1-carbon metabolism, despite its potential detrimental effect on myofibrils and, thus, explain the lower levels of sarcomere proteins and higher levels of free amino acids in the Hq heart.

Although the differences in amino acid content in the quadricep muscles of Hq mice were less pronounced than in the heart—where significance was only reached for valine, lysine and arginine, and a trend (*p* < 0.1) was observed for threonine, leucine, isoleucine, and phenylalanine−, overall, all amino acids tended to be more abundant in the Hq mice, suggesting active protein catabolism probably due, at least partly, to the observed enhanced autophagy. The higher content of free amino acids and autophagy markers in Hq quadriceps is in accordance with previous findings on the Deletor mouse muscle phenotype [48,77]. In addition, we have previously reported higher mTOR activation in the skeletal muscles of Hq mice in comparison with WT mice [41]. These data support the idea of an enhancement of 1-carbon metabolism and a protein catabolic response also in the skeletal muscles of the Hq mouse and might contribute to explaining the muscle atrophy and myopathy previously reported in this model [41].

The increased protein catabolism observed in the skeletal muscle tissue of the Hq mouse could also result in a higher availability of amino acids for other metabolic pathways in order to compensate for the mitochondrial defect. In this regard, computer models of human complex I deficiency in cardiomyocytes have predicted that the supply of several amino acids (glutamate, arginine, proline, valine, aspartate, lysine and glutamine) increase maximal ATP production by the mitochondrial respiratory chain [80]. Another study performed with transmitochondrial cybrids defective in mitochondrially-encoded ATP synthase membrane subunit 6 (MT-ATP6) and the *Cox*10^−/−^ mouse model of mitochondrial myopathy has demonstrated that anaplerosis of the Krebs cycle with α-ketoglutarate, which can be obtained from glutamine and glutamate, mitigates the consequences of the mitochondrial defect [81]. Given that skeletal muscles represent the main reservoir of amino acids in the body, we can hypothesize that muscle catabolism serves as a source of amino acids for Krebs cycle anaplerosis and energy production in the Hq mouse model, particularly the CNS. Further detailed studies are needed to assess this proposed metabolic rewiring, by analyzing whether amino acids are consumed only in the muscles, or if they are also delivered to the CNS to enhance the residual energy production capacity of this tissue.

## 5. Conclusions

AIF deficiency induces tissue-specific alterations of autophagy in the Hq mouse model of MD, with opposite changes in striated muscle tissues (heart and skeletal muscle) and in the CNS. In striated muscles, autophagy alterations may be related to metabolic rewiring through an mTOR-dependent enhancement of 1-carbon metabolism, and subsequent amino acid-driven anaplerosis of the Krebs cycle to promote energy production in other tissues, particularly the CNS. Such metabolic rewiring could, however, have a detrimental effect on muscle structure and function due to the breakdown of contractile proteins. Further research is needed to assess the precise role of autophagy disruptions in the pathophysiology of MD.

## Figures and Tables

**Figure 1 antioxidants-11-00510-f001:**
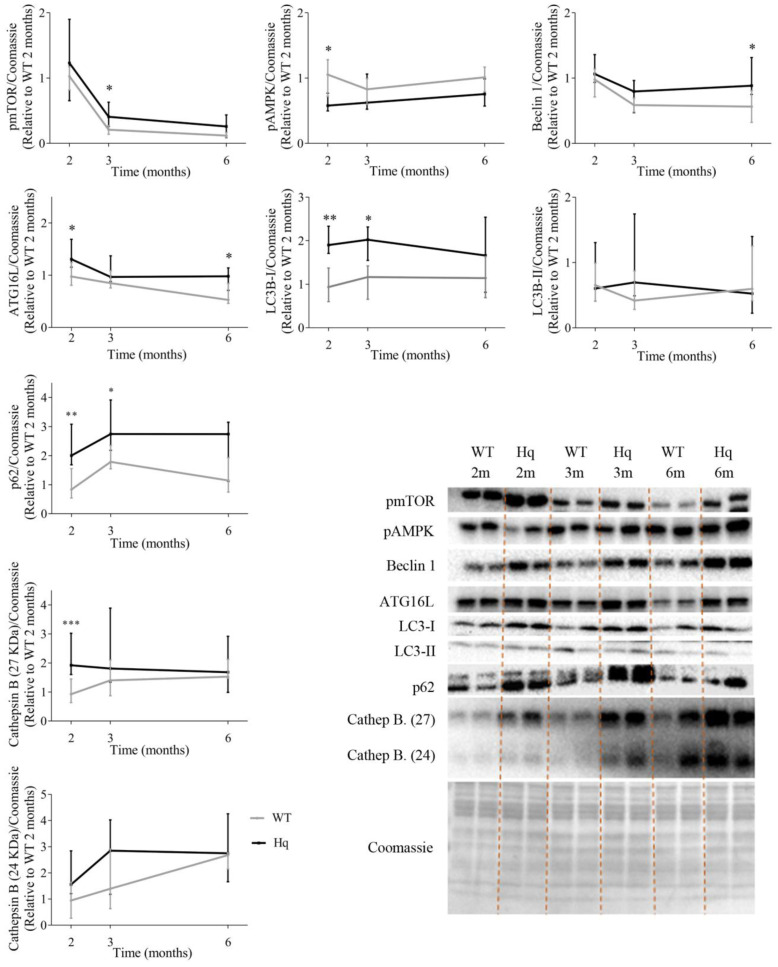
Autophagy proteins in the heart (time course study). Representative Western blots and quantifications of proteins involved in autophagy in total heart homogenates from wild-type (WT) and Harlequin (Hq) male mice at 2, 3 and 6 months (m) (*n* = 8 mice). Total protein content per lane estimated by Coomassie Blue staining was used as loading control. Data (median and interquartile range) are expressed relative to protein levels in the 2-month-old WT group. Black lines: Hq mice; gray lines: WT mice. Mann–Whitney U-test: * *p* < 0.05; ** *p* < 0.01; *** *p* < 0.001 significantly different from the age-matched WT group. Abbreviations: pmTOR, mTOR phosphorylated at residue Ser2448; pAMPK, AMPK phosphorylated at residue Thr172; ATG16L, autophagy-related protein 16-like; LC3B-I, microtubule-associated protein 1B light chain I; LC3B-II, microtubule-associated protein 1B light chain II; Cathep. B, cathepsin B.

**Figure 2 antioxidants-11-00510-f002:**
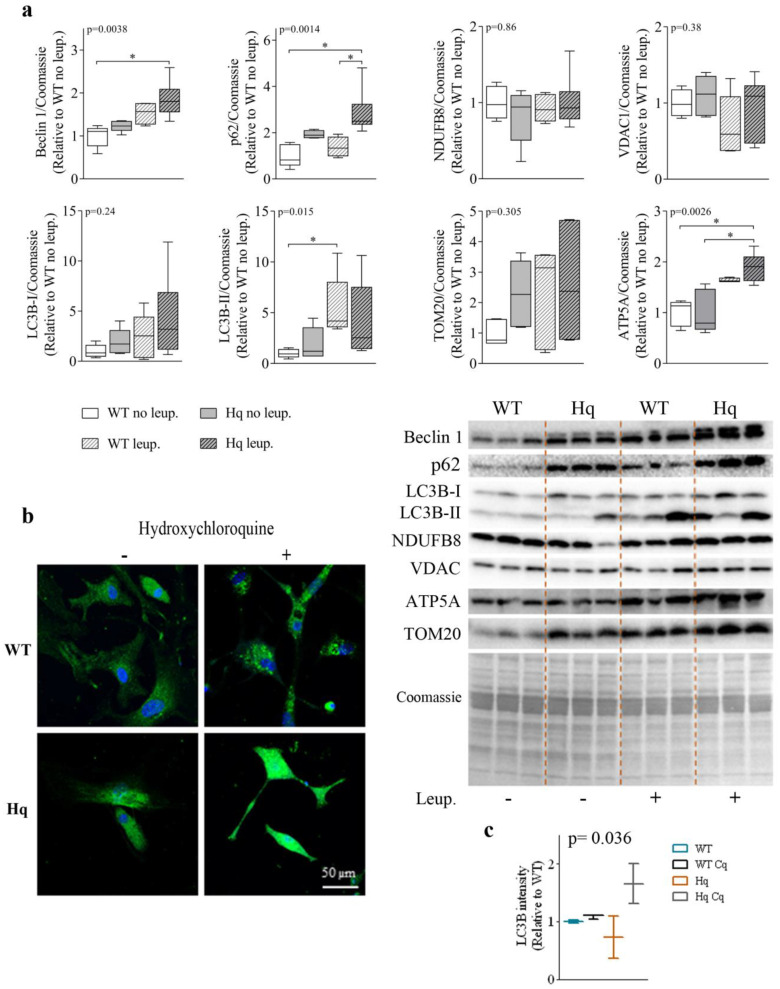
Autophagic flux and mitochondrial markers in the heart. (**a**) Representative Western blots and quantifications of autophagy (Beclin 1, p62, LC3B-I, LC3B-II) and mitochondrial (NDUFB8, VDAC, TOM20 and ATP5A) markers in heart homogenates of 3-month-old wild-type (WT) and Harlequin (Hq) male mice, whether untreated (no leup.) or treated with leupeptin (leup.) (*n* = 5–6 mice). Quantification of Western blots. Total protein content per lane estimated by Coomassie Blue staining was used as loading control. Data (median, interquartile range, minimum and maximum values) are expressed relative to the untreated WT group. (**b**) Representative images of LC3B (green) and nuclei (blue) staining in cardiomyocytes from WT and Hq male neonatal mice untreated (−) and treated (+) with hydroxychloroquine (native colors are shown). (**c**) Quantification of mean fluorescence intensity of LC3B per cell in untreated and hydroxychloroquine-treated WT and Hq neonatal cardiomyocytes (*n* = 5–15 cells from 2–3 mice). Data (median and interquartile range) are expressed relative to the untreated WT mean value. *p*-values for group effect (Kruskal–Wallis test) are shown above the graphs. Symbols for significant differences in *post hoc* (Dunn’s test) pairwise comparisons: * *p* < 0.05. Abbreviations: ATP5A, ATP synthase lipid-binding protein; TOM20, mitochondrial import receptor subunit TOM20; NDUF8, mitochondrial NADH dehydrogenase (ubiquinone) 1 beta subcomplex subunit 8; VDAC, voltage-dependent anion-selective channel protein.

**Figure 3 antioxidants-11-00510-f003:**
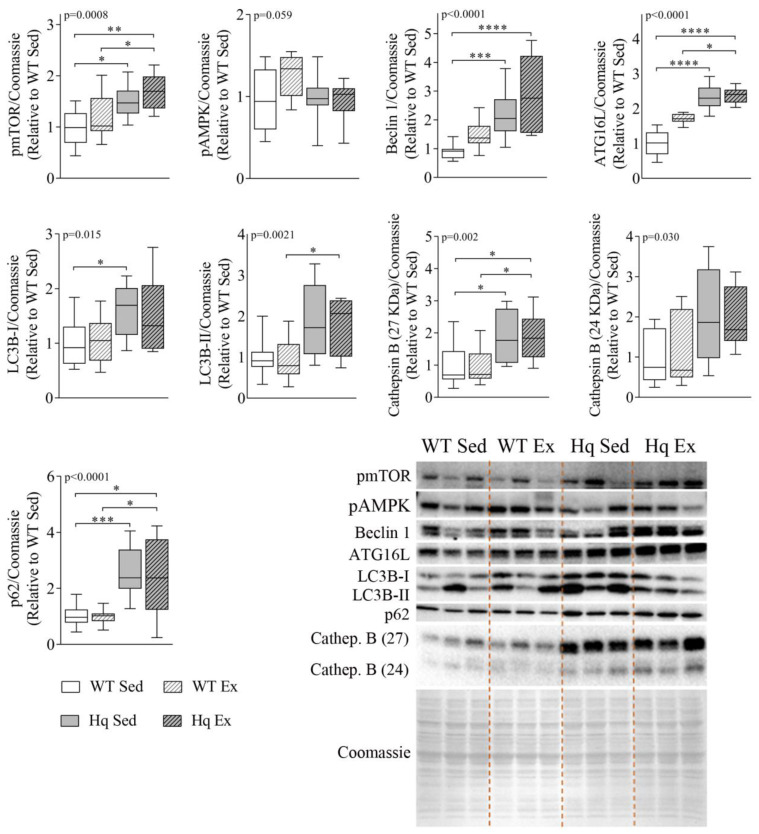
Effects of exercise training on autophagy in the heart tissue. Representative Western blots and quantifications of mTOR phosphorylated at residue Ser2448 (pmTOR), AMPK phosphorylated at residue Thr172 (pAMPK), Beclin 1, autophagy-related protein 16-like (ATG16L), microtubule-associated protein 1B light chain I (LC3B-I) and II (LC3B-II), p62 and cathepsin B (Cathep. B), in heart homogenates from wild-type (WT) and Harlequin (Hq), sedentary (Sed) and trained (Ex) mice of the intervention study (*n* = 10–12 male mice, age 5.3 months). Total protein content per lane estimated by Coomassie Blue staining was used as loading control. Data (median, interquartile range, minimum and maximum values) are relative to the control group (WT Sed). *p*-values for group effect (Kruskal–Wallis test) are shown above the graphs. Symbols for significant differences in *post hoc* (Dunn’s test) pairwise comparisons: * *p* < 0.05; ** *p* < 0.01; *** *p* < 0.001; **** *p* < 0.0001.

**Figure 4 antioxidants-11-00510-f004:**
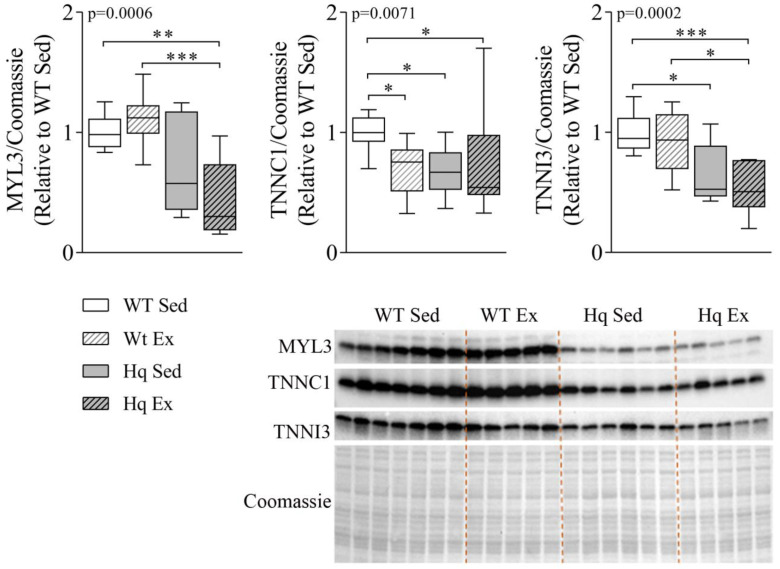
Sarcomere protein levels in the heart. Representative Western blots and quantifications of protein levels of myosin light chain 3 (MYL3), troponin C1 (TNNC1) and troponin I (TNNI3) in heart homogenates from wild-type (WT) and Harlequin (Hq), sedentary (Sed) and trained (Ex) mice in the intervention study (*n* = 10–12 male mice, age 5.3 months). Quantification of Western blots. Total protein content per lane estimated by Coomassie Blue staining was used as loading control. Significant *p*-values for group effect (Kruskal–Wallis test) are shown above the graphs. Symbols for significant differences in *post hoc* (Dunn’s test) pairwise comparisons: * *p* < 0.05; ** *p* < 0.01; *** *p* < 0.001.

**Figure 5 antioxidants-11-00510-f005:**
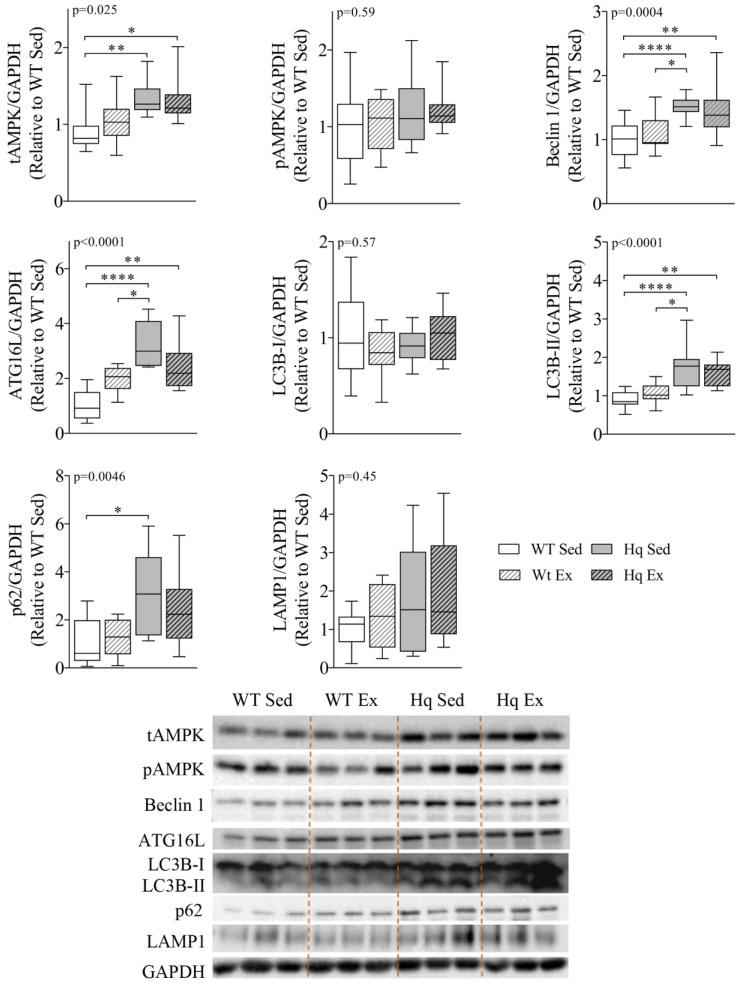
Autophagy proteins and effects of exercise training in skeletal muscle. Representative Western blots and quantifications of total AMPK (tAMPK), AMPK phosphorylated at residue Thr172 (pAMPK), autophagy-related protein 16-like (ATG16L), Beclin 1, microtubule-associated protein 1B light chain I (LC3B-I) and II (LC3B-II), p62 and lysosomal-associated membrane protein 1 (LAMP1) in *biceps femoris* homogenates from wild-type (WT) and Harlequin (Hq), sedentary (Sed) and exercise-trained (Ex) mice in the intervention study (*n* = 10–12 male mice, age 5.3 months). GAPDH was used as protein loading control. Data (median, interquartile range, minimum and maximum values) are relative to the control group (WT Sed). *p*-values for group effect (Kruskal–Wallis test) are shown above the graphs. Symbols for significant differences in *post hoc* (Dunn’s test) pairwise comparisons: * *p* < 0.05; ** *p* < 0.01; **** *p* < 0.0001.

**Figure 6 antioxidants-11-00510-f006:**
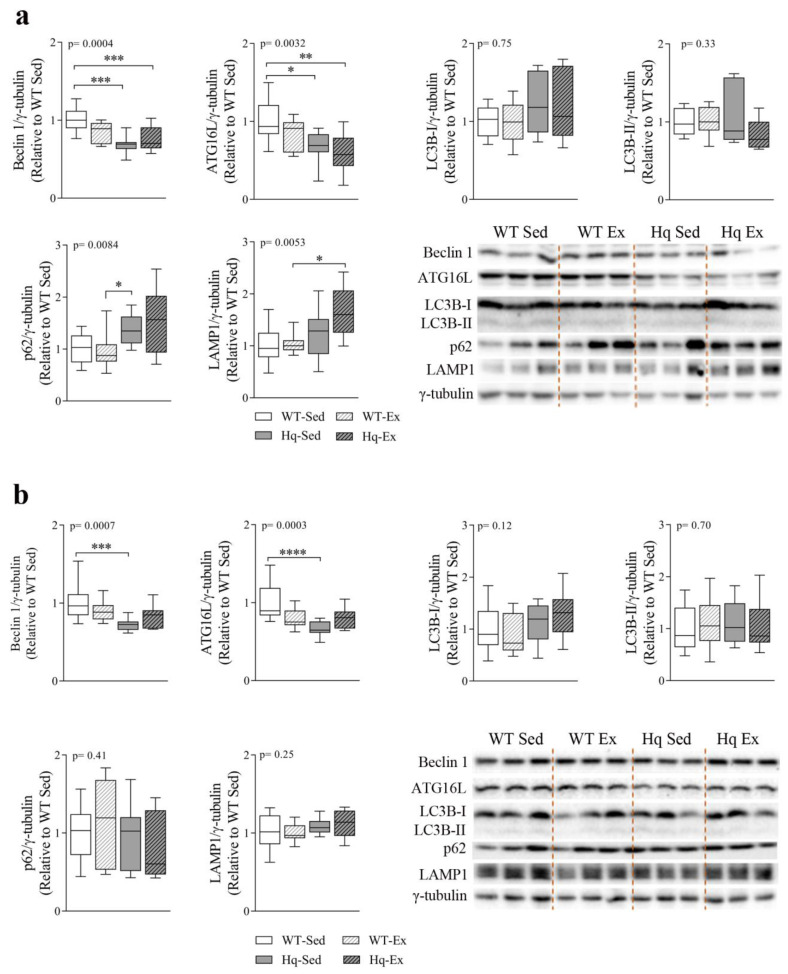
Autophagy proteins and effects of exercise training in central nervous system. Representative Western blot and densitometry analysis of autophagy-related protein 16-like (ATG16L), Beclin 1, microtubule-associated protein 1B light chain I (LC3B-I) and II (LC3B-II), p62 and lysosomal-associated membrane protein (LAMP1) in cerebellum (**a**) and brain (**b**) homogenates of wild-type (WT) and Harlequin (Hq), sedentary (Sed) and exercise-trained (Ex) mice of the intervention study (*n* = 10–12 male mice, age 5.3 months). γ-tubulin was used as protein loading control. Data (median, interquartile range, minimum and maximum values) are relative to the control group (WT Sed). *p*-values for group effect (Kruskal–Wallis test) are shown in the above graphs. Symbols for significant differences in *post hoc* (Dunn’s test) pairwise comparisons: * *p* < 0.05; ** *p* < 0.01; *** *p* < 0.001; **** *p* < 0.0001.

**Figure 7 antioxidants-11-00510-f007:**
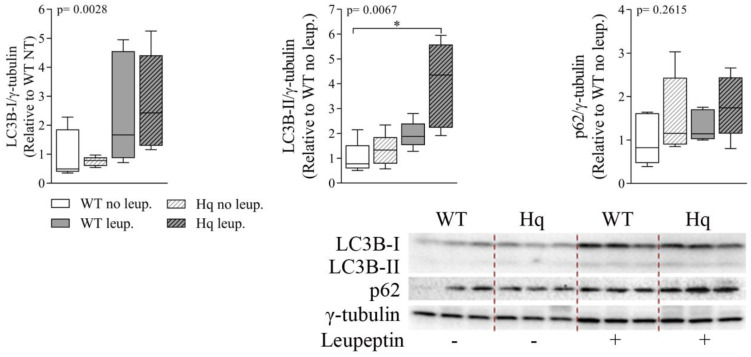
Autophagic flux analysis in the cerebellum. Representative Western blots and quantifications of autophagy markers p62, microtubule-associated protein 1B light chain I (LC3B-I) and II (LC3B-II), in cerebellum homogenates of 3-month-old wild-type (WT) and Harlequin (Hq) mice, untreated (no leup.) and treated with leupeptin (leup.) (*n* = 5–6 male mice). γ-tubulin was used as protein loading control. Data (median, interquartile range, minimum and maximum values) are expressed relative to untreated WT group. *p*-values for group effect (Kruskal–Wallis test) are shown in the above graphs. Symbols for significant differences in *post hoc* (Dunn’s test) pairwise comparisons: * *p* < 0.05.

**Table 1 antioxidants-11-00510-t001:** Free amino acids and ammonium in heart homogenates of wild-type (WT, *n* = 5) and Harlequin (Hq, *n* = 5) male mice (age 5.3 months).

Amino Acid	WT µmol/L	Hq µmol/L	*p*-Value
Threonine	14.3 [12.9–15.6]	22.3 [19.5–25.5]	0.008
Serine	17.0 [15.5–17.8]	22.4 [20.1–24.9]	0.016
Asparagine	7.9 [3.3–9.4]	9.2 [6.9–11.4]	0.309
Glutamate	192.6 [159.1–227.5]	211.4 [191.5–252.9]	0.309
Glutamine	206.7 [171.4–310.3]	294.9 [234.9–303.3]	0.309
Glycine	24.4 [20.6–26.2]	41.2 [36.6–42.8]	0.007
Alanine	131.5 [97.9–158.7]	170.5 [135.3–209.2]	0.095
Citrulline	7.0 [5.5–8.9]	7.1 [6.3–8.0]	0.690
Valine	4.8 [4.2–6.0]	10.4 [7.4–10.5]	0.031
Methionine	4.1 [2.9–4.6]	5.2 [3.7–5.5]	0.222
Isoleucine	9.0 [6.8–10.2]	11.6 [9.6–12.2]	0.151
Leucine	13.6 [8.2–13.9]	18.5 [14.1–19.1]	0.056
Tyrosine	4.9 [4.2–6.3]	6.5 [5.3–7.0]	0.309
Phenylalanine	5.8 [4.8–6.2]	5.2 [4.5–6.3]	0.690
Ammonium	313.9 [253.2–345.7]	336.9 [288.9–339.2]	>0.99
Lysine	24.4 [20.6–25.0]	27.9 [23.4–33.3]	0.095
Histidine	7.4 [6.7–10.2]	10.0 [8.6–11.3]	0.222
Arginine	16.2 [13.1–18.0]	17.8 [16.5–22.0]	0.309

Data are expressed as the median (interquartile range) of the amino acid concentration for each experimental group determined in homogenate aliquots containing 1.7 mg of total protein. *p*-values are shown for each comparison between WT and Hq groups (Mann–Whitney U-test).

**Table 2 antioxidants-11-00510-t002:** Free amino acids and ammonium in *quadriceps* homogenates of wild-type (WT, *n* = 5) and Harlequin (Hq, *n* = 5) mice (age 5.3 months).

Amino Acid	WT (µM)	Hq (µM)	*p*-Value
Threonine	38.2 [36.5–44.7]	44.7 [44.1–62.5]	0.095
Serine	47.6 [34.9–65.5]	58.7 [47.3–82.4]	0.420
Asparagine	5.6 [4.8–7.3]	6.6 [5.8–13.1]	0.548
Glutamate	56.5 [48.4–72.9]	66.2 [57.3–131.2]	0.151
Glutamine	165.3 [143.4–202.1]	204.3 [181.8–235.5]	0.095
Glycine	205.3 [170.3–311.2]	282.1 [235.1–334.4]	0.222
Alanine	348.7 [277.4–371.1]	379.4 [354.5–475.1]	0.095
Citrulline	13.6 [12.3–16.2]	15.6 [10.1–18.2]	>0.999
Valine	24.7 [23.1–25.9]	31.1 [29.3–39.1]	0.008
Methionine	9.9 [9.4–10.4]	12.1 [8.2–13.8]	0.690
Cysteine	8.3 [7.9–8.4]	8.6 [7.8–9.4]	0.151
Isoleucine	17.9 [15.8–18.8]	21.3 [18.7–23.3]	0.055
Leucine	23.6 [22.9–25.5]	33.7 [25.7–38.3]	0.095
Tyrosine	16.2 [13.2–16.3]	16.6 [14.5–18.5]	0.222
Phenylalanine	11.5 [11.2–13.1]	15.7 [11.5–17.3]	0.095
Ammonium	575.3 [573.5–600.0]	569.0 [555.1–598.5]	0.421
Ornithine	4.4 ± [4.1–5.1]	8.3 [6.2–12.8]	0.008
Lysine	53.6 [44.9–58.3]	66.8 [63.4–84.5]	0.008
Histidine	14.8 [12.9–18.7]	18.0 [14.9–21.5]	0.309
Arginine	25.0 [18.7–25.7]	34.1 [27.9–46.4]	0.008
Hydroxyproline	11.7 [10.5–14.2]	14.6 [12.3–16.1]	0.222
Proline	34.4 [16.7–37.3]	36.7 [28.9–57.8]	0.309

Data are expressed as the median ± interquartile range of the amino acid concentration (µM) for each experimental group determined in homogenate aliquots containing 2.0 mg of total protein. *p*-value is shown for each comparison between WT and Hq groups (Mann–Whitney U-test).

## Data Availability

Data is contained within the article or Appendix A.

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
