# Peer review of "Apoptosis-Inducing Factor Deficiency Induces Tissue-Specific Alterations in Autophagy: Insights from a Preclinical Model of Mitochondrial Disease and Exercise Training Effects"

_antioxidants, 2022, doi:10.3390/antiox11030510_

Round 1
Reviewer 1 Report
Manuscript by Laine-Menéndez et al. is an interesting and valuable study on the impact of AIF deficiency on autophagy status in the mouse model of mitochondrial diseases. Here are some minor comments on the manuscript:
In my opinion, the introduction should be redrafted. Currently, it is too long and some aspects are analyzed in too much detail, which can be transferred to the discussion. After reading it, I do not have the impression that I have been well introduced to the issue in question. The title of the paper indicates AIF deficiency and autophagy, MD was not directly mentioned and therefore expectations from introduction are different.
It is worth referring to MD in the title of the work, not only Harlequin mouse. This will significantly change the approach and expectations of the readers.
In my opinion, the research methodology has been described in sufficient detail, and the results are presented in a clear and structured way.
In my opinion, the research methodology has been described in sufficient detail, and the results are presented in a clear and orderly manner. The only caveat is the presentation of WB without indicating the line with the mass marker in the image. Did the authors use internal positive and negative controls for antibodies?
Data are presented as median and interquartile ranges. Was the normal distribution assessed, I did not find this information.
Finally, I congratulate the Authors a very interesting and well-written manuscript.
Author Response
Manuscript by Laine-Menéndez et al. is an interesting and valuable study on the impact of AIF deficiency on autophagy status in the mouse model of mitochondrial diseases. Here are some minor comments on the manuscript:
Comments much appreciated. We have revised the manuscript following all Reviewers’ comments (additions/changes tracked in the revised text).
Q1. In my opinion, the introduction should be redrafted. Currently, it is too long and some aspects are analyzed in too much detail, which can be transferred to the discussion. After reading it, I do not have the impression that I have been well introduced to the issue in question. The title of the paper indicates AIF deficiency and autophagy, MD was not directly mentioned and therefore expectations from introduction are different.
R1: We have now redrafted and shortened the introduction section, in which we have added a mention to the physiological role of AIF.
Q2. It is worth referring to MD in the title of the work, not only Harlequin mouse. This will significantly change the approach and expectations of the readers.
R2: We have now added ‘MD’ in the manuscript title, which has been revised in general (so that we hope it nicely illustrates now the main content of the study).
Q3. In my opinion, the research methodology has been described in sufficient detail, and the results are presented in a clear and structured way.
R3: Comment much appreciated. Thank you.
Q4. In my opinion, the research methodology has been described in sufficient detail, and the results are presented in a clear and orderly manner. The only caveat is the presentation of WB without indicating the line with the mass marker in the image. Did the authors use internal positive and negative controls for antibodies?
R4: We agree with the Reviewer that molecular weight markers help to assess if the proteins detected in western blot assays run at their predicted molecular weight position. We did not include the lines of mass markers for the sake of simplicity. Indeed, because we joined cropped images of different immunodetected proteins to create each figure panel, we would have had to include different mass marker lines for each cropped image, thereby resulting in very complicated/busy figures. In any case, the original uncropped images including the lines corresponding to mass markers of all western blots shown in the figures are available in the supplementary material of the manuscript.
Q5. Data are presented as median and interquartile ranges. Was the normal distribution assessed, I did not find this information.
R5: We actually did analyze the distribution of the data for each variable with the D’Agostino-Pearson normality test. For most of the variables analyzed, the distribution was not normal and, as such, we presented the data as median and interquartile range and used non-parametric statistical tests. This information is now included in the revised methods section.
Finally, I congratulate the Authors a very interesting and well-written manuscript.
R6. Much appreciated. Thank you!
Reviewer 2 Report
In the current manuscript, the authors Laine-Menendez S et al analyzed the alterations in cell survival mechanism autophagy and its related proteins in the different tissues includes skeletal muscle, heart, cerebellum and brain. The alterations in autophagy showed in several metabolic tissues from Harlequin (Hq) mouse, a well-established mouse model of Mitochondrial Disease (MD). This Harlequin (Hq) mouse has a markedly reduced content of AIF which has essential role in mitochondrial respiration and caspase independent cell death, providing an experimental model to query if the main role of AIF in the exacerbation of autophagy program. In addition, authors also described the influence of physical exercise intervention training on autophagy in metabolic tissues. However, there is an absence of any significant alteration in autophagy upon exercise training intervention although the authors stated in the discussion that the mice were sacrificed 48 hrs after the physical training session. Overall the current manuscript is well written and established the experimental design and results were appropriate. However, authors need to address the several comments before the manuscript is accepted in Antioxidants Journal from MDPI.
- Need to describe adequate introduction about the AIF rold and its function.
- Under normal physiological condition, how long this Hq mice can live? It is also important indicate this information in the methods or discussion. This mice had 8-week of intervention physical training and perhaps longer training period can improve the alterations in autophagy and life span of mice. It would be good if authors can comment on extended exercise training period.
- In Table 1, Free amino acids were measured in heart using HPLC methods. However, it is not mentioned that how the data was normalized? Is this data normalized to amount of tissue (dry tissue weight) used for Homogenization or protein concentration? Need to mention in the methods and Figure legend.
- Similarly, In Table 2, amino acids were measured in skeletal muscle. How the data is normalized? Need to include in methods and Figure legend.
Author Response
In the current manuscript, the authors Laine-Menendez S et al analyzed the alterations in cell survival mechanism autophagy and its related proteins in the different tissues includes skeletal muscle, heart, cerebellum and brain. The alterations in autophagy showed in several metabolic tissues from Harlequin (Hq) mouse, a well-established mouse model of Mitochondrial Disease (MD). This Harlequin (Hq) mouse has a markedly reduced content of AIF which has essential role in mitochondrial respiration and caspase independent cell death, providing an experimental model to query if the main role of AIF in the exacerbation of autophagy program. In addition, authors also described the influence of physical exercise intervention training on autophagy in metabolic tissues. However, there is an absence of any significant alteration in autophagy upon exercise training intervention although the authors stated in the discussion that the mice were sacrificed 48 hrs after the physical training session. Overall the current manuscript is well written and established the experimental design and results were appropriate. However, authors need to address the several comments before the manuscript is accepted in Antioxidants Journal from MDPI.
Comments much appreciated. Thank you. Of note, we have proofread the manuscript text in depth to improve English writing and correct spelling errors.
Q1. Need to describe adequate introduction about the AIF role and its function.
R1. Done. Please see the revised Introduction section.
Q2. Under normal physiological condition, how long this Hq mice can live? It is also important indicate this information in the methods or discussion. These mice had 8-week of intervention physical training and perhaps longer training period can improve the alterations in autophagy and life span of mice. It would be good if authors can comment on extended exercise training period.
R2. To the best of our knowledge, no study has specifically assessed the lifespan of Hq mice. Most of the studies with this mouse model have used animals aged 3-6 months, an age at which disease manifestations are clearly present. Nevertheless, the Jackson laboratory indicates that this strain shows increased sensitivity to induced morbidity/mortality, with ~30% of mice dying during the first 6 months and most deaths occurring between 15 and 30 days of age (https://www.jax.org/strain/000501). Although we have not studied the lifespan of this mouse strain (we have only worked with animals aged ≤6-7 months of age) in latter studies we have detected some premature deaths around 6 months of age in sedentary Hq mice, whereas in the present work we have not observed any premature death at this age in exercise-trained animals. However, it has been described that some Hq mice can reach 12 months of age despite strong neurodegeneration [DOI: 10.1038/nature01034]. In any case, this strain is known by its broad heterogeneity in the different disease manifestations (which by the way is also seen in MD patients) [DOI: 10.1371/journal.pone.0003208].
Concerning the comment on the training program duration, we agree that longer interventions could have maybe elicited a mitigation of the observed autophagy disturbances, but our aim in the present study was to assess if the training program could delay or mitigate the disease progression in young animals in which the disease is already present. We have now elaborated more on these issues in the revised discussion section (please see paragraph from lines 515 to 550).
Q3.In Table 1, Free amino acids were measured in heart using HPLC methods. However, it is not mentioned that how the data was normalized? Is this data normalized to amount of tissue (dry tissue weight) used for Homogenization or protein concentration? Need to mention in the methods and Figure legend.
Similarly, In Table 2, amino acids were measured in skeletal muscle. How the data is normalized? Need to include in methods and Figure legend.
R3. Free amino acid concentration was determined in aliquots of heart or skeletal muscle homogenates, respectively, using the same total amount of protein for each the two tissues (i.e., 1.7 mg for all heart homogenates and 2.0 mg for all skeletal muscle homogenates): thus, free amino acid concentration was not normalized by the amount of total protein in the sample because the latter was the same for all the heart or muscle homogenates, respectively. This has now been clarified in the revised manuscript (method section and table legends).
On behalf of all co-authors, many thanks for this insightful review.
Reviewer 3 Report
Mitochondrial diseases (MD), although being a rare diseases, constitutes serious clinical problem for patients due to alterations in the major cellular source of energy and thus mitochondrial dysfunction. From this point of view, animal model research leading to know precise molecular mechanisms of pathophysiology of MD is of high importance.
The Authors did extensive research on role of autophagy alteration in MD. I have few minor comments, I'd like to express.
- The Authors explain that one of the possible reason for absence of autophagic response to the exercise training intervention is the fact that animals were sacrificed 48h after the last physical test. Could you give rationale for suchtiming?
- Line 132. There should be numerical reference instead of referenced author’s name.
- Statistical analysis. I appreciate using non-parametric statistics. Perhaps the Authors could provide a sentence of information of the reason for choosing these tests – was it because of low number of animals in each group or non-normal data distribution?
- Figures 2-6. The explanation of what box and whiskers represent (median, interquartile range, range of the data) should be added.
- Figure 1. There is no need to provide explanation for ****, since there is no p<0.0001 in this figure.
- Figure 2b. Please, explain the readers if these colors are native or this image is pseudocolored during analysis.
- Table 1 and 2. The asterisks should be removed from the table legend, since there are no asterisks in the tables.
Author Response
Mitochondrial diseases (MD), although being a rare diseases, constitutes serious clinical problem for patients due to alterations in the major cellular source of energy and thus mitochondrial dysfunction. From this point of view, animal model research leading to know precise molecular mechanisms of pathophysiology of MD is of high importance.
The Authors did extensive research on role of autophagy alteration in MD. I have few minor comments, I'd like to express.
Q1.The Authors explain that one of the possible reason for absence of autophagic response to the exercise training intervention is the fact that animals were sacrificed 48h after the last physical test. Could you give rationale for such timing?
R1. Animals were sacrificed 48 hours after the last physical test as a means to prevent an important potential confounder as is the possible short-term effects of an acute exercise session. Indeed, we solely wanted to study the long-term effects of repeated exercise sessions, that is, training effects per se (and not the short-term effects of a single session). This has now been clarified in the materials and methods section of the revised manuscript (lines 132-136 + please see also a mention in lines 528-530).
Q2. Line 132. There should be numerical reference instead of referenced author’s name.
R2. Thank you for noting this mistake, this had now been corrected
Q3. Statistical analysis. I appreciate using non-parametric statistics. Perhaps the Authors could provide a sentence of information of the reason for choosing these tests – was it because of low number of animals in each group or non-normal data distribution?
R3. Comment appreciated. We did analyze the distribution of the data for each variable with the D’Agostino-Pearson normality test. Because for most of the variables we analyzed the data did not follow a normal distribution, we used median and interquartile range for descriptive stats as well as non-parametric tests. This information is now included in the materials and methods section of the revised version of the manuscript.
Besides, it has been documented that the use of parametric analyses for small sample sizes such as the one in our study (n ≤ 30) is not actually a correct approach [Le Boedec, K. Vet Clin Pathol 45/4: 648–656.2016].
Q4. Figure 1. There is no need to provide explanation for ****, since there is no p-value <0.0001 in this figure.
R4. This has now been removed.
Q5. Figure 2b. Please, explain the readers if these colors are native or this image is pseudocolored during analysis.
R5. Native colors obtained by LC3 staining with a primary antibody coupled to a secondary Alexa 488 secondary antibody, and DAPI staining of the nuclei are shown in Figure 2b. This point has now been clarified in the figure legend.
On behalf of all co-authors, many thanks for this insightful review.